# Drug–Drug Interactions of Cannabidiol with Standard-of-Care Chemotherapeutics

**DOI:** 10.3390/ijms24032885

**Published:** 2023-02-02

**Authors:** Tereza Buchtova, David Lukac, Zdenek Skrott, Katarina Chroma, Jiri Bartek, Martin Mistrik

**Affiliations:** 1Institute of Molecular and Translational Medicine, Faculty of Medicine and Dentistry, Palacký University, 77 147 Olomouc, Czech Republic; 2Danish Cancer Society Research Center, DK-2100 Copenhagen, Denmark; 3Department of Medical Biochemistry and Biophysics, Science for Life Laboratory, Division of Genome Biology, Karolinska Institute, 171 77 Stockholm, Sweden

**Keywords:** cannabidiol, drug–drug interaction, chemotherapy, cancer

## Abstract

Cannabidiol (CBD) is an easily accessible and affordable Marijuana (*Cannabis sativa* L.) plant derivative with an extensive history of medical use spanning thousands of years. Interest in the therapeutic potential of CBD has increased in recent years, including its anti-tumour properties in various cancer models. In addition to the direct anticancer effects of CBD, preclinical research on numerous cannabinoids, including CBD, has highlighted their potential use in: (i) attenuating chemotherapy-induced adverse effects and (ii) enhancing the efficacy of some anticancer drugs. Therefore, CBD is gaining popularity as a supportive therapy during cancer treatment, often in combination with standard-of-care cancer chemotherapeutics. However, CBD is a biologically active substance that modulates various cellular targets, thereby possibly resulting in unpredictable outcomes, especially in combinations with other medications and therapeutic modalities. In this review, we summarize the current knowledge of CBD interactions with selected anticancer chemotherapeutics, discuss the emerging mechanistic basis for the observed biological effects, and highlight both the potential benefits and risks of such combined treatments. Apart from the experimental and preclinical results, we also indicate the planned or ongoing clinical trials aiming to evaluate the impact of CBD combinations in oncology. The results of these and future trials are essential to provide better guidance for oncologists to judge the benefit-versus-risk ratio of these exciting treatment strategies. We hope that our present overview of this rapidly advancing field of biomedicine will inspire more preclinical and clinical studies to further our understanding of the underlying biology and optimize the benefits for cancer patients.

## 1. Introduction

Understanding drug interactions is a fundamental yet challenging pharmacological issue that is especially problematic in oncology due to the usually narrow therapeutic window and potential serious toxicity profiles of drugs that are often applied to vulnerable patients and those with a long history of pre-treatment. Drug interactions may occur as a result of pharmacokinetic, pharmacodynamic, or biological factors, resulting in various outcomes, including decreased or increased therapeutic or adverse responses [1]. Despite extensive research and development work, most anticancer therapies have severe adverse effects. To mitigate such therapy-associated adverse effects, patients frequently use vitamins, dietary supplements, or herbal products, with *Cannabis* plant-based products occupying a prominent position. Indeed, a recent survey indicated that approximately two-thirds of patients with cancer had used *Cannabis* products during their ongoing therapy to alleviate adverse effects [2,3]. In these studies, cannabis products were mainly taken via inhalation, ingestion, or topically. These products were predominantly in the form of edible, liquid, or smoked/vaporized cannabis. Additionally, a significant percentage of participants (22% and 21%) preferred CBD-only or -dominant products, respectively [3].

Cannabidiol (CBD), a product of *Cannabis sativa* L. (dominantly present in chemotypes II and III) [4], is particularly popular among patients with cancer because it is non-psychoactive, safe, and well-tolerated. CBD relieves therapy-induced issues, and some reports suggest it may even have direct anticancer properties. However, whether CBD improves or undermines, concomitant chemotherapy treatment has not yet been fully determined.

CBD belongs to the cannabinoid family and is a leading derivative of *Cannabis sativa* L.; CBD and tetrahydrocannabinol (THC) are the plant’s main cannabinoid constituents [5] and have been extensively studied for years due to their therapeutic potential. Unlike THC, CBD is not psychoactive and is, therefore, a more suitable therapeutic candidate.

Based on the many studies showing the benefits of CBD, historical experience with cannabinoids, and current public interest, research into CBD for medical purposes has gained increasing attention. Currently, CBD is used as a medicine or complementary substance, and many CBD-based products are publicly available. However, it should be noted that most such products are off-label and often lack CBD-content verification. Currently, CBD is available in cosmetics, beverages, edibles, solutions, herbal extracts, and dried marijuana herbs and is easily accessible worldwide [6]. The anti-inflammatory, anti-oxidative, and other biological effects of CBD make it a promising therapeutic candidate for many diseases, including epilepsy, Alzheimer’s disease, Parkinson’s disease, and multiple sclerosis. Pure CBD solutions (under the commercial name Epidiolex/Epidyolex) for oral application are currently approved by the US Food and Drug Administration (FDA) to treat severe, orphan, early-onset, and treatment-resistant epilepsy syndromes [7,8,9,10,11]. Moreover, clinical trials examining the potential benefits of CBD in the context of Parkinson’s disease, Crohn’s disease, social anxiety disorder, and schizophrenia [12,13,14,15,16] have shown promising results.

CBD-related interactions with anticancer drugs are frequently linked to the transport of chemotherapeutics due to CBD’s ability to modulate various receptors and transmembrane channels, including cannabinoid receptors 1 and 2 (CB1 and 2) [17], transient receptor potential vanilloid 1 and 2 (TRPV1 and 2) [18,19], peroxisome proliferator-activated receptor γ (PPARγ) [20], 5-hydroxytryptamine receptor subtype 1A (5-HT1A) [21], G protein-coupled receptor 55 (GPR55) [22], and adenosine A2A receptor (A2A, agonist) [23]. CBD targets have been suggested to play a role in cancer therapy resistance, including breast cancer resistance protein (ABCG2/BRCP) [24], bile salt export pump (ABCB11/BSEP) [11], and p-glycoprotein (p-gp) [25,26]. Very relevant for drug–drug interactions, CBD also affects several biotransformation enzymes such as CYP3A4, UGT1A9, UGT2B7, CYP1A2, CYP2B6, CYP2C8, CYP2C9, and CYP2C19 [11]. Several studies have furthermore examined the ability of CBD to modulate the activity of other CYPs in vitro, including 3A5/7, 2D6, 2A6, 1A1, 1B1, and 2J2 [6]. Additionally, CBD appears to have bidirectional anti-oxidative [27] and pro-oxidative properties [28], potentially modulating drug efficacy. Moreover, CBD is metabolized by CYP3A4 and CYP2C19, making it a substrate shared by multiple oncology drugs. This circumstance may slow their respective metabolic processing in the organism, thereby potentially leading to dangerous outcomes [6].

CBD is generally perceived as a well-tolerated drug, beneficial for oncology patients due to its ability to attenuate some chemotherapy-induced side effects such as nausea, pain, and appetite loss [29]. Moreover, there is evidence to suggest that CBD has direct anti-proliferative and pro-apoptotic effects while also interfering with cancer-related processes, including angiogenesis, cell migration, adhesion, and invasion [30].

In this review, we summarize the current knowledge of CBD interactions with selected anticancer chemotherapeutics and discuss emerging mechanistic explanations of their biological effects.

## 2. CBD Interactions with Antimetabolites

Antimetabolites (Table 1) are small molecules resembling standard nucleotides—the building blocks of DNA. Because of this resemblance, they can be incorrectly incorporated into DNA or may inhibit DNA synthesis. Antimetabolites were among the first anticancer chemotherapeutics introduced in the late 1940s; a notable example is aminopterin, which was used to treat paediatric acute lymphoblastic leukaemia. Currently, antimetabolites represent a cornerstone of treatment for various cancer types [31,32].

### 2.1. 5-Fluorouracil

5-fluorouracil is an uracil analogue that inhibits thymidylate synthetase. The inhibited enzyme is incapable of deoxythymidine monophosphate conversion, which results in the depletion of deoxythymidine triphosphate used for DNA synthesis [32,33]. 5-fluorouracil is commonly used to treat a range of diverse solid tumours, including those of the gastrointestinal tract and head-and-neck, with common adverse effects such as gastrointestinal and oral mucositis [32,33]. Cuba et al. (2020) [33] evaluated the effect of CBD on oral mucositis inflicted by 5-fluorouracil in mice. Interestingly, CBD treatment reduced oral mucositis, probably as a consequence of its antioxidant properties, which were documented by the increased catalase and glutathione levels in CBD-cotreated mice. CBD with 5-fluorouracil and some other chemotherapeutics is planned to be tested for efficacy and safety in a randomized clinical trial on colon and rectal cancers (NCT03607643) [34].

### 2.2. Gemcitabine

Gemcitabine is a nucleoside analogue—specifically, a deoxycytidine. After intake, the drug is metabolized and incorporated into the DNA strand instead of the original nucleoside, which leads to inhibition of DNA polymerase and ribonucleotide reductase [32,35]. Gemcitabine is commonly used in patients with non-small cell lung, pancreatic, urinary bladder, and breast cancer, with haematological toxicity, oedema, and pulmonary, cutaneous, and gastrointestinal toxicity as common adverse effects [35]. Although the impact of CBD on gemcitabine-induced adverse effects is unknown, preclinical studies suggest an interesting potentiation of its anticancer toxicity. For example, combined treatment with CBD and gemcitabine significantly extended animal survival in the pancreatic ductal adenocarcinoma (PDAC) mouse model and also enhanced the inhibition of the proliferation of PDAC cell lines. This effect can be explained by CBD’s antagonism of the GRP55 protein, a receptor of lysophosphatidylinositol, which is known to have a role in tumour progression [36,37,38]. GRP55 inhibition suppresses cell cycle progression and hence cell proliferation, reduces MAPK kinase activation, and lowers the overall abundance of ribonucleotide reductases in PDAC cells, all of which are associated with gemcitabine resistance [36,38]. The potentiation of gemcitabine toxicity by CBD was also confirmed in another PDAC cell line (MiaPaCa-2) [39]. Combined treatment with CBD and gemcitabine will be assessed for pancreatic cancer in a randomized clinical trial (NCT03607643) [34].

### 2.3. Methotrexate

Folates, compounds related to the vitamin D family, function via one-carbon metabolism, which is essential for purine, thymidylate, and polyamine synthesis. Folates are involved in the synthesis of S-adenosyl methionine, which is critical for the methylation of DNA, histones, lipids, and neurotransmitters [40]. Methotrexate is an antifolate antagonist targeting dihydrofolate reductase (DHFR), a key component of the folate pathway. Due to its structural similarity to folic acid, methotrexate can block the binding pocket of DHFR, leading to secondary inhibition of downstream enzymes in the folate pathway [32,40,41,42]. Methotrexate is used in combination with other chemotherapeutics to treat several types of cancer, including acute lymphocytic leukaemia, lymphoma, carcinomas of the breast and urinary bladder, and osteosarcoma [40,41]. Brown and Winterstein (2019) [6] suggested that CBD may interact with methotrexate treatment because it inhibits the ABCG2/BCRP transporter [24], which contributes to cellular efflux of methotrexate [43]. CBD cotreatment could thus lead to methotrexate accumulation, increasing the drug’s efficacy but also potentially strengthening its adverse effects. However, this hypothesis awaits experimental confirmation.

**Table 1 ijms-24-02885-t001:** Summary of the drug–drug interactions of CBD with antimetabolites.

Antimetabolites									
CT	Aim	Model	Administration	CBD c	CT c	Evaluation Time	Special Condition	Results of Combined Treatment	References
5-FU	Attenuation of oral mucositis	CF-1 mouse strain	IP	3, 10, 30 mg/mL daily between days 4 and 7	60 mg/kg/day on days 0 and 2	Days 8 and 11	Mechanical trauma	Reduced oral mucositis	Cuba et al. (2020) [33]
GEM	Chemotherapy efficiency	KPC mice with PDAC	IP	100 mg/kg daily	100 mg/kg every 3 days	Until death or pre-assigned endpoints are reached		Extended animal survival	Ferro et al. (2018) [36]
	Signalling pathway acquiring resistance to GEM	PDAC cell line— HPAFII and PANC1		5, 10 µM	20, 250, 500 nM	48–72 h		Decreased markers of resistance	
	Viability	PANC-1 and MiaPaCa-2		(6.25), 12.5, 25 µM daily	(25), 50, 100 µM single administration	72 h		Potentiated GEM toxicity	Luongo et al. (2020) [39]

5-FU: 5-fluorouracil; c: concentration; CBD: cannabidiol; CT: chemotherapeutics; GEM: Gemcitabine; IP: intraperitoneal; PDAC: pancreatic ductal adenocarcinoma.

## 3. Interactions of CBD with Alkylating Agents and Platinum-Based Drugs

Alkylating agents (Table 2) are reactive chemical substances that target DNA and proteins and belong to the oldest chemotherapeutics. Their ability to alter biological molecules is based on the transfer of alkyl groups to target molecules such as DNA, which often alters their structure or disrupts their function [44,45]. The alkylating agent mustard gas was first used as a chemotherapeutic in the early 1930s to treat skin cancer [45,46]. Mechlorethamine, an analogue of mustard gas, was approved as a chemotherapeutic in the late 1940s, followed by a number of related drugs [45].

Platinum-based drugs are commonly classified as alkylating agents even though they do not directly alkylate DNA; instead, they form covalent crosslinks between DNA strands and covalent DNA-protein adducts [45].

### 3.1. Carmustine

Carmustine (1,3-bis(2-chloroethyl)urea) belongs to the nitrosourea family. As a lipid-soluble, low molecular weight drug, it readily penetrates the blood–brain barrier and is therefore used to treat brain tumours, mainly glioblastoma multiforme as well as lymphomas and melanoma [45,47,48]. Reported adverse effects associated with carmustine include pulmonary, gastrointestinal, hepatic, renal, cardiovascular, haematological, and neurological toxicities, among others [48].

It has been hypothesized that CBD interferes with carmustine via the transient potential vanilloid 2 receptor (TRPV2), a nonselective cation channel that is usually activated by heat, altered osmolarity, or membrane stretching. As a TRPV2 agonist, CBD increases Ca^2+^ influx [19], possibly affecting the uptake and retention of some anticancer drugs. This theory was experimentally confirmed in glioblastoma cells, where CBD coadministration caused sensitization against carmustine. Interestingly, this effect was not observed in normal human astrocytes [49,50]. TRPV2 expression is also upregulated by a spliced variant of the AML1a (acute myeloid leukaemia) protein, induced by CBD treatment. As a transcription factor, AML1a impacts the expression of genes whose products are implicated in various biological processes, including hematopoietic self-renewal, cell proliferation, and differentiation [51,52]. Although the function of AML1a in GBM (glioblastoma multiforme) remains poorly understood, AML1a is upregulated during GBM stem-like cell (GSC) differentiation, and downregulation of AML1a is associated with GBM chemoresistance. Thus, upregulation of AML1a by CBD may lead to GSC differentiation, resulting in restoration of carmustine sensitivity. This concept was experimentally validated at the level of GSC cell lines [52]. The potentiating effect of CBD on carmustine cytotoxicity was further examined by Deng et al. (2017) [53] in human GBMs, mouse primary GBMs, and mouse neural progenitor cells. Although CBD sensitized to carmustin both the murine and human GBM cells, a subsequent analysis of drug interactions between carmustine and CBD, their concentration dependence, and their effects on efficacy revealed a range of concentration-dependent synergistic, additive, or antagonistic effects on cell viability and proliferation among the three analysed model systems, overall indicating the complexity of such drug interactions [53].

### 3.2. Temozolomide

Temozolomide is an imidazotetrazine lipophilic prodrug that is activated and metabolized under physiological pH conditions. The resulting methyldiazonium salt is a lipophilic alkylating agent that passes through the blood–brain barrier. Consequently, temozolomide is particularly suitable for treating brain tumours [54,55]. Reported adverse effects of this drug include haematological, gastrointestinal, and neurological toxicities [56]. Experiments using the human U87MG glioma cell line, primary glioblastoma cells, and normal human astrocytes (NHAs) conducted by Nabissi et al. (2013) [49] suggested that temozolomide’s efficacy is potentiated by CBD, again via TRPV2 activation. As in the case of carmustine, temozolomide cytotoxicity was potentiated in glioblastoma cells but not in NHAs.

The CBD-promoted potentiation of temozolomide’s effects against GBM may also partly reflect effects on extracellular vesicles (EVs). Cells use EVs for drug efflux, pro-oncogenic signalling, invasion, and immunosuppression [57], and there is clear evidence that CBD is an effective modulator of EV properties. For example, CBD treatment of chemoresistant and chemosensitive patient-derived GBM cells led to the formation of EVs with reduced levels of pro-oncogenic miR21 and an elevated anti-oncogenic miR126 [58]. The same study suggested that, apart from its effects on EV properties, CBD treatment may reduce levels of prohibitin, a protein that protects mitochondrial function and may contribute to temozolomide chemoresistance.

In addition to the additive/synergistic effects observed between CBD and temozolomide toxicity, concentration-dependent antagonistic effects have been reported in human GBMs, mouse primary GBM cells, and mouse neural progenitor cells, suggesting a rather complex relationship [53]. In one study, the combination of CBD with temozolomide increased tumour growth in a mouse GBM model compared to temozolomide treatment alone [59]. However, the opposite effect was observed in another study using the same mouse model [60]. It is unclear why these two studies reached such highly discrepant results.

Notably, there are also limited clinical data. Likar et al. (2019) [61] evaluated brain tumours in patients who took CBD concomitantly with radiochemotherapy, including temozolomide. At the time of publication, patients taking CBD survived longer than expected. Currently, two clinical trials evaluate the effects of CBD and temozolomide combinations in patients with GBM (NCT03607643 and NCT03687034) [34,62].

### 3.3. Cisplatin

Cisplatin ((SP-4-2)-diamminedichloridoplatinum (II)) interacts with purines incorporated into DNA and causes DNA lesions. It is widely used as a chemotherapeutic agent for treating solid tumours despite having severe side effects, including renal toxicity, ototoxicity, hepatotoxicity, and gastrointestinal toxicity [45,63]. Several lines of evidence indicate that CBD is an effective attenuator of these cisplatin-induced adverse effects. For example, renal toxicity, a frequent limiting factor for cisplatin treatment, was effectively suppressed by CBD in mice [64]. Indeed, CBD was shown to protect kidneys from cisplatin-induced effects by reducing oxidative and nitrosative stress, inflammation, and cell death, while improving renal function. Moreover, studies on cisplatin-induced emesis in shrews reported by Kwiatkowska et al. (2004) [65] showed that treatment with low doses of CBD attenuated cisplatin-induced vomiting, whereas high doses potentiated this adverse effect. Similar findings were obtained in another study on shrews by Rock et al. (2012) [66], which in a mouse model showed that the anti-emetic and anti-nausea effect of CBD was mediated by indirect activation of the somatodendritic serotonin autoreceptor 5-HT1A in the dorsal raphe nucleus. Activation of 5-HT1A reduces the excitability of serotoninergic neurons and thus reduces serotonin release in terminal forebrain regions [66,67]. The same mechanism was confirmed in shrews and rats after treatment with cannabidiolic acid (CBDA), a compound similar to CBD that is effective at substantially lower concentrations [68].

Besides the ability to reduce adverse effects, CBD also seems to potentiate cisplatin toxicity in endometrial cancer cell lines. The enhanced response to cisplatin appeared to result from TRPV2 receptor activation and was strengthened in TRPV2 overexpression models [69]. Conversely, in the ovarian cancer cell line SKOV-3, CBD did not affect cisplatin-induced cytotoxicity, and a cell protective effect was observed at higher CBD concentrations [70]. Similarly, in GBM cell lines, combined treatment with CBD and cisplatin had mixed results; additive, synergistic, and antagonistic effects were all observed, depending on the drug concentrations used [53].

### 3.4. Oxaliplatin

Oxaliplatin is a third-generation platinum drug based on diaminocyclohexane (DACH) that can overcome cisplatin-induced resistance. Oxaliplatin is less reactive toward DNA than cisplatin, forming fewer protein-DNA adducts and DNA-DNA crosslinks. Despite its lower reactivity, oxaliplatin displays a similar cytotoxicity profile to cisplatin. It is believed that the DACH molecule causes specific DNA lesions that are difficult to detect and/or repair. Moreover, oxaliplatin interacts more strongly with proteins than cisplatin [71,72]. In addition to generating DNA lesions, oxaliplatin induces nucleolar and ribosomal stress, which may contribute significantly to its overall toxic effects [73]. Oxaliplatin is primarily used to treat colon cancer [63,72] and has less severe side effects than cisplatin; nephrotoxicity and ototoxicity are relatively rare. Moreover, while oxaliplatin treatment can cause gastrointestinal toxicity and hepatotoxicity, the main dose-limiting adverse effect is neurotoxicity [72] accompanied by mechanical allodynia. Importantly, CBD is a potent attenuator of oxaliplatin-induced neuropathy-associated pain in mice [74]. Pereira et al. (2021) [75] recently investigated the role of the endocannabinoid system in oxaliplatin-induced peripheral sensory neuropathy and showed that CBD has an analgesic effect that was partly attributed to interactions with the CB1 receptor. CBD treatment also attenuated mechanical hyperalgesia and c-Fos expression in the spinal cord’s dorsal root ganglion and dorsal horn without cannabimimetic effects.

In another study on the potentiation of oxaliplatin by CBD, experiments using parental and oxaliplatin-resistant human colorectal cancer cell lines and their mouse xenografts conducted by Jeong et al. (2019) [76] showed that CBD restored oxaliplatin sensitivity tested in the resistant cancer cells. Mechanistically, this was attributed to a CBD-promoted decrease in NOS3 phosphorylation, which is otherwise elevated in oxaliplatin-resistant tumour cell lines. Reducing NOS3 phosphorylation lowered the level of NO and superoxide dismutase 2, resulting in ROS induction and mitochondrial dysfunction. Oxaliplatin combined with CBD is under evaluation in a randomized clinical trial of colorectal cancer (NCT03607643) [34].

### 3.5. Carboplatin

Carboplatin, or cis-diamine-1,1′-cyclobutane dicarboxylate platinum (II), is a second-generation platinum-based drug. Its mechanism of action resembles that of cisplatin, and its therapeutic effect is weaker than or equal to that of cisplatin itself [77,78]. Carboplatin is used to treat testicular germ cell tumours as well as gynaecological, head-and-neck, thoracic, and urinary bladder cancers [79]. It has fewer adverse effects than cisplatin because of its different pharmacodynamics; its main adverse effects are myelosuppression, nausea, and vomiting [80]. Current knowledge of the interaction between CBD and carboplatin is limited because the combination of both drugs has only been tested in cellular models of canine urothelial carcinoma, in which the effect was antagonistic [81].

**Table 2 ijms-24-02885-t002:** Summary of the drug–drug interactions of CBD with alkylating agents and platinum-based drugs.

Alkylating Agents and Platinum-Based Drugs									
CT	Aim	Model	Administration	CBD c	CT c	Evaluation Time	Special Condition	Results of Combined Treatment	References
BCNU	Viability	U87MG, MZC, NHA		10 µM	10^−5^–10^−3^ M/200 µM	24, 72 h		Increased toxicity, except for NHA	Nabissi et al. (2013) [49]
	Colony formation	U87MG, MZC		10 µM	200 µM	Day 14		Decreased colony formation	
	Apoptosis	U87MG, MZC		10 µM	200 µM	6 h		Increased annexin	
	TRPV2 function	U87MG, MZC		10 µM	200 µM	Day 1		TRPV2 dependent	
	Cell viability, differentiation, apoptosis, mitochondrial activity	GCS lines of patients with cancer		10 µM	200 µM	24 h	Medium + EGF and bFGF	Restoration of BCNU sensitivity	Nabissi et al. (2015) [52]
	Proliferation, viability	GBM (Hu, Ms), Ms NPCL		0.3–100 µM	3 µM to 1 mM	72 h		Concentration-dependent effect	Deng et al. (2017) [53]
TMZ	Viability	U87MG, MZC, NHA		10 µM	10^−5^–10^−2^ M/400 µM	24, 72 h		Increased toxicity except for NHA	Nabissi et al. (2013) [49]
	Colony formation	U87MG, MZC		10 µM	400 µM	Day 14		Decreased colony formation	
	Apoptosis	U87MG, MZC		10 µM	400 µM	6 h		Increased annexin	
	TRPV2 function	U87MG, MZC		10 µM	400 µM	Day 1		TRPV2 dependent	
	EV release, miRs, prohibitin	LN18, LN229		5 µM	800 µM	1 h		Anti-oncogenic effect	Kosgodage et al. (2019) [58]
	Proliferation and viability (effect of CBD up to BCNU toxicity)	GBM (Hu, Ms) + Ms NPCL		0.3–100 µM	1 µM to 1 mM	72 h		Concentration-dependent effect	Deng et al. (2017) [53]
	Survival	Patients with brain cancer	Capsule (CBD)	100 mg twice daily, increased up to 200 mg twice daily	Standard therapy		Surgical resection + radiotherapy	Prolonged life	Likar et al. (2019) [61]
	Tumour volume	Nude mice with U87 MG	Oral (CBD), IP (TMZ)	15 mg/kg/day	5 mg/kg/day	Day 15		Increased tumour growth	López-Valero et al. (2018) [59]
	Viability	Patient-derived GBM cells and four Glioma cell lines (U251, U87 MG, LN18)		10, 20, 30 µM	200, 500 µM	48 h		Synergic effect	Huang et al. (2021) [60]
	Growth inhibition	U251, LN18, and GL261 sphere culture		30 µM	200 µM	24, 48 h		Synergic effect	
	Colony formation assay	U251, U87 MG		20 µM	500 µM	Day 7		Synergic effect	
	Autophagy markers, mitophagy induction (U251)	U251, U87 MG		30 µM	500 µM	24, 48 h		Increased autophagy and mitophagy	
	Tumour growth, survival, markers of autophagy, mitophagy, and proliferation	Nude mice with U87 MG-GFP-luc	IP	15 mg/kg/once daily	25 mg/kg/once daily	Days 7, 14, 21, and 28		Decreased tumour growth	
CDDP	Renal function	Male C57BL/6J mice	IP	(2.5, 5), 10 mg/kg 1.5 before (or 12 h after) CDDP, daily	20 mg/kg single administration	72 (+ 1.5) h		Decreased renal toxicity	Pan et al. (2009) [64]
	Histopathological damage, ROS production, apoptosis, inflammation response, nitrosative stress	Male C57BL/6J mice	IP	10 mg/kg/day 1.5 h before CDDP	20 mg/kg single administration	72 (+ 1.5) h		Decreased renal toxicity	
	CDDP-induced vomiting	Shrews	IP	5 (attenuation), 40 (potentiation) mg/kg 0.5 h before CDDP treatment	20 mg/kg	1 h observation	Mealworms 15 min before pre-treatment	Modulation according to the concentration of CBD	Kwiatkowska et al. (2004) [65]
	CDDP-induced vomiting	Shrews	SC (CBD), IP (CDDP)	5, 10 mg/kg 30 min before CDDP	20, (40) mg/kg	1 h observation	Mealworms 15 min before pre-treatment	Anti-emetic and anti-nausea effect	Rock et al. (2012) [66]
	CDDP-induced vomiting	Shrews	IP	CBCA: 0.5 mg/kg 45 min before CDDP	20 mg/kg	70 min observation	mealworms 15 min before pre-treatment	Attenuation of vomiting	Bolognini et al. (2013) [68]
	Viability	Ishikawa		3.92 µg/ml	0.25, 0.5 µg/ml	72 h		Increased CDDP toxicity	Marinelli et al. (2020) [69]
	Viability	SKOV-3		1, 10 µM pre-treatment for 24 h; 10, 15, 20 µM co-treatment	5–100 µM	(24 +) 48 h		No effect (or antagonistic effect)	Fraguas-Sánchez et al. (2020) [70]
	Proliferation, viability	GBM (Hu, Ms) + Ms NPCL		0.3–100 µM	0.1–100 µM	72 h		Concentration-dependent effect	Deng et al. (2017) [53]
L-OHP	Mechanical allodynia	Male C57Bl6 mice	IP	1.25–10 mg/kg 15 min before L-OHP	6 mg/kg single administration	Days 2, 4, 7, and 10		Attenuation of mechanical allodynia	King et al. (2017) [74]
	Chemotherapy efficiency—viability, cell death, autophagy, ROS, oxygen concentration, mitochondrial function	colo205 R, DLD-1 R		4 µM	10 µM	6, 12, 24 h		Sensitization of resistant cells	Jeong et al. (2019) [76]
	Tumour growth, autophagy	Female BALB/c nude mice with colo205 R	IP	10 mg/kg every 2 days	5 mg/kg every 2 days	Day 18		Lower tumour weight	
	Peripheral sensory neuropathy	Swiss male mice	PO (CBD), IV (L-OHP)	10 mg/kg, 3 times/week 1 h before L-OHP or in mid-term between L-OHP injections	2 mg/kg twice a week	Days 28 and 56	Mechanical hyperalgesia—the tip of a rigid filament 1 week before drug injection, repeated once a week. Cold allodynia—tail immersed in cold water, once a week, 120 s cut-off time	Attenuation of peripheral sensory neuropathy	Pereira et al. (2021) [75]
CBDCA	Viability, combination index, apoptosis	AXA, Orig, and SH cell lines		0.03–300 μM; IC50: 5.77, 5.30, and 5.48 μM (and derived concentration series)	0.01–1 mM; IC50: 384, 529, and 398 μM (and derived concertation series)	24 h	0.1% FBS	Antagonistic effect	Inkol et al. (2021) [81]

BCNU: carmustine; bFGF: basic fibroblast growth factor; CBD: cannabidiol; CBDCA: carboplatin; CBDA: cannabidiolic acid; CDDP: cisplatin; c: concentration; CT: chemotherapeutics; EGF: epithelial growth factor; EV: extracellular vesicles; GBM: glioblastoma; GSC: glioblastoma stem-like cells; Hu: human; IP: intraperitoneal; IV: intravenous; L-OHP: oxaliplatin; Ms: mouse; NHA: normal human astrocytes; NPCL: neural progenitor cell line; PO: per os; ROS: reactive oxygen species; SC: subcutaneous; TRPV2: transient receptor potential vanilloid 2; TMZ: temozolomide.

## 4. Interactions of CBD with Microtubule-Targeting Agents

Microtubules (Table 3) are components of the cytoskeleton that are dynamically polymerized and depolymerized. Microtubule-targeting agents inhibit one of these two processes, which has a profound impact on cytoskeleton morphology, cellular transport, motility, and mitosis, and thereby exert anticancer and anti-angiogenic effects [82,83]. The first microtubule-targeting agent used in the clinic was vinblastine, which was approved for the treatment of lymphomas and various solid tumours in the early 1960s. Its success led to the discovery and development of a wider spectrum of agents with similar activity, with several of these drugs currently used to treat various types of malignancies [83].

### 4.1. Vinblastine

Vinblastine is an indole-containing alkaloid isolated from *Catharanthus roseus* and its endophytes [84] that shows microtubule-destabilizing properties. As an inhibitor of microtubule polymerization [82], it is widely used to treat Hodgkin’s lymphoma, lymphosarcoma, choriocarcinoma, neuroblastoma, various carcinomas, leukaemia, Wilkins’s tumour, and reticulum cell sarcoma [85]. Its known adverse effects include gastrointestinal, genitourinary, neurologic, haematological, and hepatic toxicities and patients treated with vinblastine are also prone to infections [86].

The effect of CBD on vinblastine-induced adverse effects is not yet known, but some studies suggest potentiation of the curative effect. For example, in leukaemia cells, CBD can help overcome vinblastine resistance that is caused by p-glycoprotein (p-gp) transporter overexpression. P-gp overexpression is responsible for multidrug resistance, and vinblastine is among its known substrates. CBD was shown to reduce the transporter’s expression, reducing the efflux of vinblastine from cells. CBD cotreatment also increased vinblastine toxicity, but only in cells overexpressing the transporter. Notably, the effect was relatively mild, and verapamil, an established inhibitor of the p-gp transporter, was considerably more effective [25]. The synergistic effect of CBD and vinblastine manifested by reduced viability and increased apoptosis was further confirmed in canine urothelial carcinoma cells [81].

### 4.2. Paclitaxel

Paclitaxel, a member of the taxane family isolated from *Taxus brevifolia* [87], is a microtubule-stabilizing agent used to treat various solid tumours. Its common adverse effects include myelosuppression and peripheral neuropathy [88]. Ward et al. (2011; 2014) [89,90] used CBD to attenuate paclitaxel-induced peripheral neuropathy and showed that CBD cotreatment prevented this side effect in mice [89,90]. The neuroprotective activity of CBD was partially explained by its agonistic effect on the 5-HT1A receptor, which is involved in central nervous system sensitization [89,90,91]. The same study also reported additive and synergistic effects in murine and human breast cancer cell lines [90]. The attenuating effect on paclitaxel-induced mechanical allodynia was also confirmed by King et al. (2017) [74] in mice. Apart from 5-HT1A, the protective effect of CBD on paclitaxel-induced peripheral neuropathy might reflect the impact on mitochondrial Na^+^ Ca^2+^ exchanger-1 (mNCX-1), which is primarily responsible for calcium homeostasis. Indeed, the knockdown of mNCX-1 attenuated the protection conferred by CBD in dissociated dorsal root ganglia [92]. Another study confirmed CBD as an attenuator of paclitaxel-induced mechanical sensitivity and showed that CBD could not reverse previously established mechanical sensitivity in mice [93].

Other studies have examined the capacity of CBD to modulate the efficacy of paclitaxel. Synergistic or additive effects with CBD pre-/co-treatment on cytotoxicity were observed in breast cancer and ovarian cancer cell lines. These studies tested two CBD preparations—CBD in solution and CBD in polymeric microparticles—both of which were effective in the human MCF7, MDA-MB-231, and SKOV-3 cell lines and reduced tumour growth during in ovo studies on the chorioallantoic membrane of chicken embryos [70,94].

According to Luongo et al. (2020) [39], CBD and paclitaxel cotreatment had synergistic and additive effects in pancreatic cancer cell lines but only at relatively high CBD concentrations. Synergy was also observed in the MCF7 breast cancer cell line model [95]. CBD and paclitaxel cotreatment has also been investigated in human colorectal and gastric adenocarcinoma cell lines [96]. While CBD did not reduce viability after paclitaxel treatment in some cell lines, an additive effect on the inhibition of DNA replication was observed. Interestingly, the authors highlighted the importance of the foetal calf serum content in cell culture media, leading to different results after CBD application, suggesting the potential influence of growth factors on the observed effects of CBD. In addition, CBD enhanced the cytotoxic effect of paclitaxel in endometrial cancer cell lines regardless of TRPV2 overexpression [69].

Finally, Brown and Winterstein (2019) [6] demonstrated that paclitaxel is a substrate of the ABCB11/BSEP transporter, which is another CBD target. Combined treatment with CBD may thus increase the efficacy of paclitaxel, albeit at the cost of a simultaneous increase in adverse effects.

### 4.3. Docetaxel

Docetaxel is a semisynthetic agent consisting of 10-diacetyl baccatin III derived from *Taxus baccata* that has been esterified with a synthetic side chain. Docetaxel is a microtubule stabilizer [97] that is used to treat metastatic breast, lung, prostate, gastric, and head-and-neck cancers; it seems to be more potent than paclitaxel. However, its poor solubility in water necessitates administration as a solution in ethanol and polysorbate 80, which introduces additional solvent-induced side effects. Docetaxel also has more severe adverse effects than paclitaxel, including neutropenia, musculoskeletal toxicity, and neurotoxicity [98].

De Petrocellis et al. (2013) [99] studied the pro-apoptotic effects of cannabinoids when combined with docetaxel in prostate carcinoma. Both CBD and CBD-DBS (extracts from *Cannabis sativa* L. strains enriched in particular cannabinoids) were examined. Interestingly, these researchers highlighted the importance of media composition (with/without sera) in experiments using CBD by showing that certain media can potentiate docetaxel toxicity in human LNCaP (androgen receptor AR-positive) cells. In the same study, CBD also potentiated the effect of docetaxel in AR-negative human DU-145 cells at lower concentrations but showed a tendency towards a protective effect at higher concentrations. In experiments using the DU-145 mouse xenograft model, CBD-BDS potentiated the effects of docetaxel, but a mild protective effect was observed following cotreatment with CBD and docetaxel in an LNCaP xenograft model. Combined treatment with docetaxel and CBD has also been examined in the MCF7 breast adenocarcinoma cell line, revealing that synergistic effects can be obtained at specific molar ratios [95].

### 4.4. Vincristine

Vincristine is a vinca alkaloid isolated from *Catharanthus roseus* that interferes with microtubule polymerization [100] and is used to treat cancers including leukaemia, lymphoma, central nervous system cancers, and sarcomas. Its common adverse effects include neuropathy and constipation [101].

While mechanical allodynia, a neurological condition induced by vincristine, was not attenuated by CBD pre-treatment [74], multiple lines of evidence suggest that CBD potentiates the anticancer effect of vincristine. First, as mentioned above, CBD targets the ATP-binding cassette (ABC) transporter (ABCC1/MRP1) responsible for multidrug resistance, including resistance to vincristine. In ovarian cancer cell lines overexpressing ABCC1/MRP1, CBD treatment attenuated the ABCC1/MRP1-mediated drug transport and increased the accumulation of vincristine inside the cells [102]. Second, combined CBD and vincristine treatment reduced cell proliferation in a synergic or additive manner in canine neoplastic cell lines. The authors suggested that this effect could be related to CBD-mediated induction of ERK and JNK kinase-mediated phosphorylation signalling leading to autophagy and apoptosis [103]. Third, a case report suggested a favourable impact of CBD on patients with high-grade gliomas treated with radiotherapy accompanied by a combination of vincristine, lomustine, and procarabine. Importantly, two patients cotreated with CBD showed an improved health condition that exceeded expectations [104].

**Table 3 ijms-24-02885-t003:** Summary of the drug–drug interactions of CBD with microtubule-targeting agents.

Microtubule-Targeting Agents									
CT	Aim	Model	Administration	CBD c	CT c	Evaluation Time	Special Condition	Results of Combined Treatment	References
VBT	Viability	CCRF-CEM, CEM/ VLB100		10 µM	0.1 nM to 10 µM	72 h		Increased toxicity	Holland et al. (2006) [25]
	Viability, combination index, apoptosis	AXA, Orig, and SH cell lines		0.03–300 μM; IC50: 5.77; 5.30, and 5.48μM (and derived concentration series)	0.01–10 μM; IC50: 2.51; 2.23; and 3.09 μM (and derived concertation series)	24 h	0.1% FBS	Increased toxicity	Inkol et al. (2021) [81]
PTX	Cold and mechanical allodynia	C57Bl/6 mice female and male	IP	5 or 10 mg/kg daily on days 1–14	1, 2, 4 or 8 mg/kg on days 1, 3, 5, and 7	Testing every 3–10 day (for 66 days)	Cold allodynia—acetone; mechanical allodynia—von Frey filaments	Attenuation of cold and mechanical allodynia	Ward et al. (2011) [89]
	Mechanical allodynia	female C57Bl/6 mice	IP	2.5–10 mg/kg 15 min before PTX	4, 8 mg/kg on days 1, 3, 5 and 7	Weekly for 10 weeks	von Frey monofilaments	Attenuation of mechanical allodynia	Ward et al. (2014) [90]
	Viability	LN 231, 4T1		1–4 µM	2,5–35 µM	48 h		Increased toxicity	
	Mechanical allodynia	Male C57Bl/6 mice	IP	0.625–20 mg/kg 15 min before PTX	8 mg/kg on days 1, 3, 5, and 7	Reassessed on days 9, 14, and 21		attenuation of mechanical allodynia	King et al. (2017) [74]
	CBD-mediated protection against PTX	Dissociated DRG from embryonic rats		10 µM	3 µM	5 h		Knockdown of Mncx-1 attenuated CBD-mediated protection against PTX	Brenneman et al. (2019) [92]
	Viability, pre-administration strategy	MCF-7, MDA-MB-231		2.5, 5 and 10 µM (MCF-7); 1.25, 2.5 and 5 µM (MDA-MB-231) 24 h before PTX	10–500 nM	24 + 48 h		Increased toxicity	Fraguas-Sánches et al. (2020) [94]
	Viability, co-treatment strategy	MCF-7, MDA-MB-231		10, 15 and 20 µM (MCF-7); 5, 7.5 and 10 µM (MDA-MB-231)	10–500 nM	48 h		Increased toxicity	
	Viability, combination studies	MCF-7, MDA-MB-231		CBD in solution 5 or 10 µM daily; CBD-Mps (single administration), started 24 h before PTX	10–500 nM	24 + 48 h		Increased toxicity with both formulations	
	Tumour growth	MDA-MB-231 grafted onto CAM membrane	Topically	CBD in solution 100 µM daily, CBD-Mps ones (single administration) 24 h before PTX	100 µM	24 + 48 h		Reduced tumour growth	
	Viability, pre-administration study	SKOV-3		1 and 10 µM for 24 h before PTX	10–500 nM	24 + 48 h		Increased toxicity with 10 µM CBD	Fraguas-Sánches et al. (2020) [70]
	Viability, coadministration study	SKOV-3		10, 15, and 20 µM	10–500 nM	48 h		Increased toxicity	
	Viability, pre- and coadministration study	SKOV-3		CBD in solution 10 µM daily; CBD-Mps (single administration), started 24 h before paclitaxel	10–500 nM	24 + 48 h		Increased toxicity (Mps are more effective)	
	Tumour growth	SKOV-3	Topically	CBD in solution 100 µM daily, CBD-Mps once (single administration) 24 before PTX	100 µM	24 + 36 h		Reduced tumour growth	
	Viability	PANC-1 and MiaPaCa-2		6.25, 12.5, 25 µM	1.75, 3.5, 7 µM	72 h		Increased toxicity	Luongo et al. (2020) [39]
	Viability and synergy study	MCF7		CBD:PTX (1:9, 2:8, 3:7, 4:6, 5:5, 6:4, 7:3, 8:2, and 9:1, *v*/*v*)		72 h		Found the most synergistic ration	Alsherbiny et al. (2021) [95]
	Apoptosis and necrosis	MCF7		64.6 µM	0.1 µM	24 h		Enhanced cell deaths (CBD toxic itself)	
	Viability, DNA synthesis	HT29		0.5–10 µM	10 nM	72 h		No effect	Sainz-Cort et al. (2020) [96]
	Viability, DNA synthesis	AGS, SW480		0.5–10 µM	2 and 10 nM	72 h		No effect at viability, increased inhibition of DNA synthesis	
	Viability	Ishikawa		3.92 µg/ml	0.0015 and 0.003 µg/ml	72 h		Increased toxicity	Marinelli et al. (2020) [69]
	Mechanical sensitivity	Male C57Bl/6	IP	2.5 mg/kg on days 1, 3, 5 and 7; 15 min before PTX	8 mg/kg on days 1, 3, 5 and 7	Days −3, −2, −1, and 14	von Frey monofilaments	Prevention against the development of mechanical sensitivity	Foss et al. (2021) [93]
	Mechanical sensitivity	Male C57Bl/6	PO, IP	0.25, 2.5, 25 mg/kg on days 1, 3, 5 and 7; 15 min before PTX	8 mg/kg on days 1, 3, 5 and 7	Days −3, −2, −1, and 14	von Frey monofilaments	Prevention against the development of mechanical sensitivity	
	Mechanical sensitivity	Male C57Bl/6	IP	20 mg/kg on days 12, 13 and 14	8 mg/kg on days 1, 3, 5 and 7	Days −3, −2, −1, 11, and 14	von Frey monofilaments	CBD did not reverse mechanical sensitivity	
DTX	Xenograft growth	Male MF-1 nude mice	IP+IV	100 mg/kg CBD-BDS daily	5 mg/kg once weekly	4–5 weeks observation		Different results according to xenograft origin	De Petrocellis et al. (2013) [99]
	Viability and proliferation	LNCaP, DU-145		1–25 µM	increasing concentration	72 h		Effect modulated by CBD concentration and sera presence	
	Viability and synergy study	MCF7		CBD:DTX (1:9, 2:8, 3:7, 4:6, 5:5, 6:4, 7:3, 8:2, and 9:1, *v*/*v*)		72 h		Found the most synergistic ration	Alsherbiny et al. (2021) [95]
	Apoptosis, necrosis	MCF7		39.75 µM	0.5 µM	24 h		Increased apoptosis and necrosis	
VCT	Mechanical allodynia	Male C57Bl6 mice	IP	1.25–10 mg/kg 15 min before VCT	0.1 mg/kg daily for 7 days	Days 5, 10, 15, and 22		No effect	King et al. (2017) [74]
	VCT accumulation	Hu ovarian carcinoma cell line 2008/MRP1		2–100 µM 30 min before VCT	100 nM	30 + 90 min	Absence of serum	Increased VCT intracellular concentration	Holland et al. (2008) [102]
	Viability	Canine neoplastic cell lines		0.34, 0.67, 1.25, 2.5, 5, 10, 20 g/ml	0.25–6.6 nM	48 h		Reduced cell proliferation	Henry et al. (2021) [103]
	Disease progression	Patients with high-grade glioma		100–450 mg/day	Standard PCV therapy		Surgical resection + radiotherapy	Improved health condition	Dall’Stella et al. (2018) [104]

c: concentration; CAM: chicken chorioallantoic membrane; CBD: cannabidiol; CT: chemotherapeutics; DRG: dorsal root ganglion; DTX: docetaxel; Hu: human; IP: intraperitoneal; IV: intravenous; Mps: microparticles; PO: per os; PTX: paclitaxel; VBT: vinblastine; VCT: vincristine.

## 5. CBD Interactions with Anthracyclines

The anthracyclines (Table 4) are a class of compounds that intercalate into DNA, causing inhibition of DNA synthesis and interference with topoisomerase II [105]. The proto-typic anthracyclines, doxorubicin and daunorubicin, were isolated from *Streptomyces peucetius* in the 1960s. Since then, several other compounds of this class have received clinical approval [106].

### Doxorubicin

Doxorubicin has many therapeutic applications, including in the treatment of haematological malignancies, diverse types of carcinomas, and sarcomas. Its main adverse effects include cardiotoxicity, neurotoxicity, nephrotoxicity, and hepatotoxicity [107]. Doxorubicin has a complex molecular mechanism of action that involves both free radical induction and DNA intercalation leading to DNA breaks and chromosomal aberrations, often resulting in cell senescence or death [108].

Cardiomyopathy is the most severe doxorubicin-evoked adverse effect, which is manifested by myocardial injury, oxidative and nitrative stress, cardiac dysfunction, reduced mitochondrial biogenesis and function and decreased expression of medium-chain acyl-CoA dehydrogenase and uncoupling proteins 2 and 3 [109]. Preclinical studies have indicated that CBD may effectively attenuate these effects. For example, Fouad et al. (2013) [110] examined the effects of CBD on doxorubicin-mediated cardiotoxicity in rats, revealing that rats co-treated with CBD had attenuated cardiac injury, along with reduced levels of serum creatine kinase-MB, troponin T, lipid peroxidation, cardiac malondialdehyde, tumour necrosis factor-α, nitric oxide, calcium ions, nitric oxide synthase, nuclear factor-κB, Fas ligand, and caspase-3. These CBD-mediated effects were accompanied by increased levels of cardiac glutathione (GSH), selenium, zinc ions, and survivin expression in cardiac tissue, all features that likely contributed to improved histological and biomarker readouts. A cardioprotective effect of CBD in doxorubicin-treated mice was also reported by Hao et al. (2015) [109], who observed reduced levels of cardiac oxidative and nitrative stress markers, inflammation, and cell death together with enhanced expression of matrix metalloproteinases and improvements in cardiac biogenesis and mitochondrial function.

The potentiation of doxorubicin’s anticancer activity by CBD has also been widely studied. Increased doxorubicin accumulation was observed after CBD treatment of Caco-2 and LLC-PK1/MDR1 cancer cell lines; this effect is probably attributable to the blocking of doxorubicin efflux by CBD-induced inhibition of the p-gp transporter [26]. CBD is also an agonist of the TRPV2 channel, which increases doxorubicin uptake, as shown in the U87MG glioma cell line [49] and hepatocellular carcinoma cell lines [111]. Positive modulation of TRPV2 by CBD and improved efficacy of doxorubicin were also confirmed in human triple-negative breast cancer cell lines, including their mouse xenografts [112]. Consistently, CBD enhanced the cytotoxic effect of doxorubicin in an endometrial cancer cell line overexpressing TRPV2 to a greater extent than was observed in parental cells with regular TRPV2 expression [69]. Synergic effects of CBD and doxorubicin were also confirmed in the MCF7 breast adenocarcinoma cell line [95].

Patel et al. (2021) [113] tested combined treatment with doxorubicin and free CBD or CBD encapsulated in EVs as a new delivery system in a model of triple-negative breast cancer (TNBC). Both formulations showed comparable potency in terms of sensitizing cancer cells to doxorubicin, which was manifested by the accumulation of cells in the G1 phase, and affected markers of inflammation, metastasis, and apoptosis. In the xenograft model, the combined treatment led to a reduction in tumour volume that was comparable for both CBD formulations.

CBD formulated in microparticles has also been tested with doxorubicin as a pre-/co-administration strategy, with several preclinical studies showing an enhanced cytotoxic effect in human breast cancer and ovarian cell lines, but only when using a pre-plus cotreatment strategy [70,94].

CBD-mediated potentiation of doxorubicin efficacy has recently been investigated in two TNBC cell lines [114]. The authors observed variable effects (without pre-treatment) as CBD cotreatment resulted in synergism or antagonism depending on the cell line and drug concentration used. Moreover, CBD was shown to effectively sensitize cells against doxorubicin in 3D cultures. Combining the two substances also potentiated the anti-migratory effect in human TNBC MDA-MB-231 cells. The authors suggested that this effect may reflect mechanisms involving integrins, LOX, ATG5, autophagy and ABCA2, all of which are downregulated by CBD.

Combined CBD and doxorubicin treatment have also been tested for potential veterinary cancer therapy applications, suggesting that they synergistically or additively reduce cell proliferation in canine neoplastic cell lines [103]. Additionally, an antagonistic effect was described for lower concentrations of CBD and doxorubicin [103].

**Table 4 ijms-24-02885-t004:** Summary of the drug–drug interactions of CBD with anthracyclines.

Anthracyclines									
CT	Aim	Model	Administration	CBD c	CT c	Evaluation Time	Special Condition	Results of Combined Treatment	References
DOX	Cardiomyopathy	Male Sprague-Dawley rats	IP	5 mg/kg/day for 4 weeks	2.5 mg/kg 6x every 48 h for 2 weeks	4 weeks + 1 day		Attenuation of cardiomyopathy	Fouad et al. (2013) [110]
	Cardiomyopathy	Male C57BL/6J mice	IP	10 mg/kg 1.5 h before DOX and daily	20 mg/kg	5 days		Attenuation of cardiomyopathy	Hao et al. (2015) [109]
	Drug accumulation	Caco-2 cells		1, 3, 10, 30 µM	1 µM	1 h		Increased drug accumulation	Zhu et al. (2006) [26]
	Drug accumulation	LLC-PK1 and LLC-PK1/MDR1		5, 20, 100 µM	1 µM	1 h		Increased drug accumulation	
	Viability	U87MG, MZC, NHA		10 µM	10^–5^–10^–3^ M	24, 72 h		Increased toxicity except for NHA	Nabissi et al. (2013) [49]
	Colony formation	U87MG, MZC		10 µM	200 µM	14 days		Decreased colony formation	
	Apoptosis	U87MG, MZC		10 µM	200 µM	6 h		Increased annexin	
	DOX uptake	MZC		10 µM 30 min before DOX	5 µM	0.5 + 2 h		TRPV2 dependent	
	TRPV2 function	U87MG, MZC		10 µM	200 µM	1 day		TRPV2 dependent	
	TRPV2 function	Murine BNL1 ME A.7R.1 cells		10 µM co- treatment and after DOX washout	1 µM	Seconds		TRPV2 dependent	Neumann-Raizel et al. (2019) [111]
	p-gp inhibition, viability, colony formation	Murine BNL1 ME A.7R.1 cells		10 µM	0.1 µM	24 h		Increased toxicity	
	DOX uptake	SUM159 and MDA-MB231		5 µM 2 h before DOX	5 µM	2 + 0.5 h		Higher DOX uptake	Elbaz et al. (2018) [112]
	Viability	SUM159 and MDA-MB232		5 µM	0.025–64 µM	24 h		Increased toxicity	
	Apoptosis	SUM159		5 µM	0.5 µM	24 h		Increased apoptosis	
	Colony formation	SUM159 and MDA-MB232		5 µM	0.5 µM	6 days	Reduced serum	Decreased colony formation	
	Tumour growth/apoptosis	Female NU/NU nude mice with SUM159 xenograft	PT CBD; IP DOX	5 mg/kg once per week 2 h before DOX	5 mg/kg	4 weeks		Lower tumour volume, increased pro-apoptotic markers	
	Viability	Ishikawa		3.92 µg/ml	0.015 and 0.03 µg/ml	72 h		Increased toxicity	Marinelli et al. (2020) [69]
	Viability, pre-administration strategy	MCF-7, MDA-MB-231		2.5, 5 and 10 µM (MCF-7); 1.25, 2.5 and 5 µM (MDA-MB-231) 24 h before DOX	0.1–20 µM	24 + 48 h		Increased toxicity (more in MDA-MB-231)	Fraguas-Sánches et al. (2020) [94]
	Viability, co-treatment strategy	MCF-7, MDA-MB-231		10, 15 and 20 µM (MCF-7); 5, 7.5 and 10 µM (MDA-MB-231)	0.1–20 µM	48 h		Increased toxicity (except 10 µM CBD in MCF7)	
	Viability, combination studies	MCF-7, MDA-MB-231		CBD in solution 5 or 10 µM daily; CBD-Mps (single administration), started 24 h before DOX	0.1–20 µM	24 + 48 h		Increased toxicity with both formulations	
	Viability, pre-administration study	SKOV-3		1 and 10 µM for 24 h before DOX	1–60 µM	24 + 48 h		Not statistically significant	Fraguas-Sánches et al. (2020) [70]
	Viability, coadministration study	SKOV-3		10, 15, and 20 µM	1–120 µM	48 h		Not statistically significant	
	Viability, pre- and coadministration study	SKOV-3		CBD in solution 10 µM daily; CBD-Mps (single administration), started 24 h before DOX	0.1–20 µM	24 + 48 h		Increased toxicity	
	Viability	MDA-MB-231		CBD and CBD EV 1 µM (24 h before DOX)	0.156–10 µM	24 + 48 h		Increased sensitivity	Patel et al. (2021) [113]
	Cell cycle, apoptosis, inflammatory, and metastatic markers	MDA-MB-231		CBD and CBD EV 1 µM (24 h before DOX)	500 nM	24 + 48 h		Increased G1 and apoptosis, decreased inflammation and metastasis	
	Cell migration	MDA-MB-231		CBD EV 1 µM	500 nM	40 h (reading every 10 min)		Decreased migration	
	Tumour volume, apoptosis, inflammatory and metastatic markers	Envigo nude mice (MDA-MB-231)	IP (CBD and CBD EV), IV (DOX)	CBD, CBD EV 5 mg/kg (1 day before DOX; twice weekly)	2 mg/kg (twice weekly)	Days 1, 4, 10, and 14		Lower tumour volume, increased apoptosis, decreased inflammation, and metastasis	
	Viability and synergy study	MCF7		CBD:DOX (1:9, 2:8, 3:7, 4:6, 5:5, 6:4, 7:3, 8:2, and 9:1, *v*/*v*)		72 h		Found the most synergistic ration	Alsherbiny et al. (2021) [95]
	Apoptosis, necrosis	MCF7		38, 42 µM	0.2 µM	24 h		Increased apoptosis and necrosis	
	Viability	Canine neoplastic cell lines		0.34, 0.67, 1.25, 2.5, 5, 10, 20 g/ml	0.033–2 µM	48 h		Reduced cell proliferation	Henry et al. (2021) [103]
	Viability, combinatorial effect	MDA-MB-231, MDA-MB-468		1, 2.5 (2D); 5 (3D) µM	0.39–25 µM (2D); 5–100 µM (3D)	24 + 48h; 48h		Increased toxicity	Surapaneni et al. (2022) [114]
	Cell migration	MDA-MB-231		1 µM	500 nM	40 h (reading every 10 min)		Anti-migratory effect	
	Immunoblotting	MDA-MB-468		1 µM	1 µM	24 + 48 h		Increased cell sensitivity against DOX	

c: concentration; CBD: cannabidiol; CT: chemotherapeutics; D: dimensional; DOX: doxorubicin; EV: extracellular vesicles; IP: intraperitoneal; IV: intravenous; Mps: microparticles; NHA: normal human astrocytes; PT: peritumoural.

## 6. Interactions of CBD with Proteotoxic Stress-Inducing Drugs

Protein homeostasis involves protein synthesis, folding, and degradation in cells. Protein turnover is generally high in cancer cells due to uncontrolled cell divisions and growth as well as numerous genetic alterations that may give rise to proteins with altered structures or alter the composition of multimeric complexes [115]. Most cancer cells, therefore, experience elevated proteotoxic stress making them highly dependent on the proper function of protein-degradation machinery such as the ubiquitin-proteasome system (UPS). Protein homeostasis is therefore seen as a promising target for cancer therapy and has been studied extensively [116]. The first drug targeting this process, bortezomib, was approved for clinical use in 2003 [117].

### 6.1. Bortezomib

Bortezomib (Table 5) is one of the clinically used proteasome inhibitors and is commonly used to treat multiple myeloma (MM). It has a broad spectrum of adverse effects, including haematological and gastrointestinal toxicity and neurotoxicity [118]. CBD was shown to synergistically potentiate the anticancer effect of bortezomib, leading to increased growth inhibition, cell cycle arrest, and cell death through the ERK/AKT/NFκB pathway in human MM cell lines. The strongest cytotoxic response to bortezomib in combination with CBD was observed in TRPV2-overexpressing MM cells [119]. Bortezomib combined with CBD is about to be tested in phase II clinical trial of patients diagnosed with MM, GBM, and GI malignancies (NCT03607643) [34].

### 6.2. Disulfiram

Proteasome inhibitors aside, there are currently no approved anticancer drugs that directly target the UPS or protein homeostasis. However, some FDA-approved medicines targeting UPS that were initially approved for other indications are now being repurposed for cancer treatment. Disulfiram, which has been used for over 60 years to treat alcoholism, is an ideal candidate for repurposing as it is a well-tolerated drug with a substantial anticancer effect supported by numerous preclinical studies, case reports, and clinical trials, including several ongoing clinical trials (NCT04521335; NCT03323346; NCT03950830) [120,121,122,123,124]. The UPS-targeting effect of disulfiram is due to its metabolite, a copper-diethyldithiocarbamate complex (CuET), which targets the NPL4 protein, a cofactor of p97 segregase [125]. P97, with its cofactors, is a vital component of the UPS acting upstream of the proteasome through p97’s ATPase activity that enables the segregation from diverse subcellular structures, unfolding, and translocation of ubiquitinated proteins for proteasome-mediated degradation [126]. CuET causes NPL4 aggregation leading to p97/NPL4 complex malfunction, triggering suprathreshold, irresolvable proteotoxic stress and consequently cancer cell death [125]. We recently showed that CBD effectively blocks CuET-mediated proteotoxic stress and toxicity in cancer cells by upregulating metallothioneins MT-1E and MT-2A, small proteins that are responsible for metal homeostasis and heavy metal detoxification, including copper. Because CuET is a copper complex, increased metallothionein expression reduces its activity against its primary target, NPL4, and thus reduces its toxicity [127]. These results indicate that the promising anticancer effect of the disulfiram/copper combination may be severely compromised by the simultaneous use of CBD products as supportive care.

**Table 5 ijms-24-02885-t005:** Summary of the drug–drug interactions of CBD with proteotoxic stress inductors and topoisomerase inhibitors.

Proteotoxic Stress Inductors									
CT	Aim	Model	Administration	CBD c	CT c	Evaluation Time	Special Condition	Results of Combined Treatment	References
BRT	Cytotoxicity, proliferation, cell cycle, necrosis, cell death	RPMI8226, U266		20 µM	3 ng/ml	72 h		Synergistic activity	Morelli et al. (2014) [119]
	Mitochondrial activity	RPMI8226		20 µM	3 ng/ml	1 h		Induced mitochondrial-dependent necrosis	
	ROS	RPMI8226		20 µM	3 ng/ml	2 h		Increased ROS	
DSF/CuET	Viability	U2OS, MDA-MB-231		10 µM 24 h pre-treatment + 10 µM co-treatment	62.5–500 nM	24 + 72 h		Decreased sensitivity	Buchtova et al. (2021) [127]
	Proteotoxic stress markers, MTs induction	U2OS, MDA-MB-231, RPE-2		10 µM 24 h before CuET	0.2 µM	24 + 3 h		Decreased proteotoxic stress, increased MT expression	
**Topoisomerase inhibitors**									
**CT**	**Aim**	**Model**	**Administration**	**CBD c**	**CT c**	**Time point**	**Special condition**	**Results of combined treatment**	**References**
TPC	Viability	MEF3.8		10 µM 1 h before TPC	1 nM–10 µM	1 + 48 h		Increased toxicity	Holland et al. (2007) [24]

c: concentration; CBD: cannabidiol; CT: chemotherapeutics; DSF: disulfiram; BRT: bortezomib; TPC: topotecan.

## 7. CBD Interactions with Topoisomerase Inhibitors

Topoisomerases are essential modulators of DNA replication, recombination, repair, and transcription because of their role in releasing topological DNA stress. Topoisomerases catalyse single- or double-strand cleavage of DNA, and their proper function is crucial for cellular differentiation, proliferation, and survival. These enzymes were first identified as potential anticancer targets in the mid-1980s, and several topoisomerase inhibitors have since then been discovered and established in oncological praxis [128].

### Topotecan

Topotecan (Table 5) selectively inhibits topoisomerase I, which catalyses single-strand DNA cleavage. Topotecan binds to the enzyme and prevents the re-ligation step, causing the formation of a single-strand DNA gap and simultaneously trapping the enzyme [129]. Topotecan is used as part of second-line therapy for metastatic ovarian cancers and relapsed small cell lung cancers. The most limiting adverse effect of topotecan is haematological toxicity; less severe effects include fatigue, nausea, vomiting, hypokalaemia, and increased γ-glutamyltransferase activity [130].

While the impact of CBD on adverse effects of topotecan is unknown, it seems that CBD may overcome resistance to topotecan mediated by the ABCG2/BCRP efflux transporter, which is directly targeted by CBD, as has been shown in the ABCG2/BCRP transporter-overexpressing MEF3.8 cell line [24].

## 8. Discussion

Currently, up to two-thirds of oncology practitioners say that they have discussed *Cannabis* product use with their patients but acknowledge that they do not have sufficient information to provide solid recommendations [131]. Indeed, recent surveys suggest considerable use of CBD and *Cannabis*-derived products among patients with cancer [2,3]. This review aims to provide oncologists and cancer patients with an overview of the potential benefits and drawbacks, as well as some uncertainties, associated with concomitant CBD use during ongoing chemotherapy. We have focused on summarizing the available preclinical data concerning the drug–drug interactions of CBD with commonly used chemotherapeutics in cell cultures and animal models (Table 1, Table 2, Table 3, Table 4 and Table 5). However, the list of anticancer chemotherapeutics discussed herein is not exhaustive, and the effects of CBD in combination with other cannabinoids, which are discussed in several publications, were excluded due to the length restrictions and the focus of our present review.

Patients have two primary motivations for using CBD during ongoing anticancer therapy: (i) attenuation of adverse effects and (ii) enhancement of the therapeutic efficacy. Adverse effects are often limiting factors of chemotherapy treatment, and CBD, in combination with certain chemotherapeutics, can enable patients to withstand therapy for a longer time and/or at a higher dosage. According to available surveys [2,3], adverse effects are the main reason why patients with cancer use *Cannabis* products. Among the notable preclinical examples discussed in this review, CBD can alleviate 5-fluorouracil-induced oral mucositis, cisplatin-induced renal and gastric toxicities, oxaliplatin-induced neuropathic pain, paclitaxel-induced neuropathy, and doxorubicin-induced cardiotoxicity, as well as show some desirable anti-emetic and anti-nausea effects during chemotherapy. There is a consensus in most of the cited studies that CBD has good potential to improve the quality of life of patients with cancer undergoing standard chemotherapy.

The modulation of treatment efficacy by CBD is another fascinating issue that warrants further investigation. CBD has been proven to have a multi-target impact on various cellular processes, some of which are necessary for cancer-cell survival or modulate the toxic effects of various anticancer drugs. We collected and highlighted preclinical data illustrating these impacts. For example, CBD was shown to potentiate the effects of gemcitabine, carmustine, cisplatin, temozolomide, paclitaxel, and vincristine in various preclinical models. The mechanistic explanations for these combinatorial effects mainly involve effects on drug influx/efflux resulting from interactions with transporters and/or channels such as TRPV2, which is activated by CBD and increases the influx and retention of carmustine, cisplatin, temozolomide, doxorubicin, and bortezomib. The well-known p-gp efflux transporter, which is responsible for multidrug resistance, is yet another very relevant target of CBD in this context. CBD-mediated inhibition of the p-gp transporter leads to increased accumulation of vinblastine and doxorubicin in p-gp-overexpressing cell lines. CBD also inhibits the ABCC1/MRP1 and ABCG2/BCRP transporters and thereby facilitates the accumulation of methotrexate, vincristine, and topotecan, at least in cell lines overexpressing these transporters. Importantly, the ability of CBD to increase the accumulation of these drugs is not always cancer cell-/tissue-specific, a challenging aspect that raises the risk of unwanted increased adverse chemotherapy effects and overdose in patients using CBD in conjunction with these anticancer agents.

Other modes of chemotherapy potentiation by CBD that does not involve transporters have also been proposed. These include regulation of EV trafficking of anti- and pro-oncogenic miRNAs in temozolomide-treated glioblastoma [56]; induction of ERK and JNK kinase pathways, which promotes autophagy in vincristine- and doxorubicin-treated canine neoplastic cells [103]; GRP55 inhibition, which reduces growth and cell cycle progression in gemcitabine-treated PDAC cells [36]; and ROS induction in oxaliplatin-treated colorectal cell lines [76].

Importantly, there is also preclinical evidence that CBD might significantly reduce the efficacy of anticancer drugs in some cases. This serious issue remains relatively unexplored and has been mentioned in some studies as an unexpected finding without being actively studied mechanistically. However, one mechanistic explanation has been provided recently when it was shown that CBD efficiently protects cancer cells against certain metal-containing drugs by inducing the expression of specific detoxifying proteins known as metallothioneins [127]. Indeed, metallothioneins are involved in cellular defence against multiple chemicals, and their effects are not limited to heavy metal complexes [132,133]. Consequently, this effect could be relevant for numerous anticancer drugs. Accordingly, antagonistic effects of CBD have been reported for carmustine, temozolomide, cisplatin, carboplatin, and doxorubicin. This suggests that concomitant use of CBD during anticancer therapy may be unsuitable for some drug combinations because it may appear to reduce adverse effects while actually reducing the likelihood of successful treatment.

Overall, based on the collected data here, it can be concluded that CBD shows exciting potential for improving cancer chemotherapy outcomes when combined with various standard-of-care drugs and not only in terms of side effect attenuation. For example, cancers with developed treatment resistance might benefit from concomitant CBD application due to its reported effects on transporters. In addition, combinations including carmustine, paclitaxel, and doxorubicin seem promising. However, some of the data are less conclusive, with conflicting findings involving, for example, temozolomide and cisplatin. Finally, CBD combined with carboplatin or disulfiram appears to be a potentially dangerous combination (decreasing the drug efficacy). The last two examples even raise an essential concern that some of the CBD-promoted side effect attenuation might reflect a reduced amount of available active drugs. Thus, given the lack of clinical data and only scheduled or ongoing clinical trials, the reported promising preclinical results need to be carefully evaluated and translated into meaningful clinical benefits.

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
