# Peer review of "Drug–Drug Interactions of Cannabidiol with Standard-of-Care Chemotherapeutics"

_ijms, 2023, doi:10.3390/ijms24032885_

Round 1
Reviewer 1 Report
Article Review: "Drug-drug interactions of cannabidiol with standard-of-care chemotherapeutics".
1. Cannabidiol (CBD) was previously studied to show anticancer activities. Therefore, it is logic to have synergistic action with other anticancer drugs in safe dose.
2. "Cannabidiol (CBD), a product of the Cannabis plant". Give the name of the species of cannabis that contains CBD.
3. "two-thirds of patients with cancer had used Cannabis products". Which products: herbal or isolated drugs such as CBD?.
4. In the introduction: It is not clear what is in the present actually used in the therapy: herbal remedy or the bioactive compound CBD.
5. In the tables, it is better to add column with result or type of the drug-drug interaction.
Author Response
A point-by-point response
Article: Drug-drug interactions of cannabidiol with standard-of-care chemotherapeutics (ijms-2171652)
We value the time the reviewers spent reading our manuscript and providing their valuable feedback. While revising the manuscript based on the referees' guidance, we meticulously addressed all their remarks and concerns, resulting in a better and clearer text. We hope that the reviewers will now approve the resulting revised manuscript. The point-by-point responses are given below. The yellow highlights in the revised manuscript indicate the changes. Our answers are included below.
Response to Reviewer 1
- Cannabidiol (CBD) was previously studied to show anticancer activities. Therefore, it is logic to have synergistic action with other anticancer drugs in safe dose.
We agree with the reviewer that logic would indeed suggest that combining cannabidiol (CBD) and some cytostatics could improve anticancer therapy. However, the bioactivity profile of cannabinoids is extensive and involves so many pathways that this assumption might not always be valid and requires more research and deeper scientific insight. That is why we wrote this review, and we even highlighted a few examples suggesting that the combined therapy might interfere with the treatment outcome. With our article, we hopefully provide a thought-provoking overview that would motivate further work in this rapidly evolving area of biomedicine, including more preclinical studies as well as clinical trials addressing the so-far insufficiently understood interactions of cannabinoids with standard-of-care chemotherapy.
- "Cannabidiol (CBD), a product of the Cannabis plant". Give the name of the species of cannabis that contains CBD.
Thank you - the information has been included (see line: 48-49) in the revised manuscript.
- "two-thirds of patients with cancer had used Cannabis products". Which products: herbal or isolated drugs such as CBD?.
Thank you again for the question. This information has now been added to the text. (see lines: 44-47 in the revised manuscript)
- In the Introduction: It is not clear what is in the present actually used in the therapy: herbal remedy or the bioactive compound CBD.
This point has been partially addressed in the response to point 3 above (see lines: 44-47 in the revised manuscript). We also include the official commercial name of the FDA-approved pure CBD (line:67)
- In the tables, it is better to add column with result or type of the drug-drug interaction
This is a good point, thank you. We have added such a column (see the revised tables 1-5)
Reviewer 2 Report
The authors have discussed CBD interactions with selected anticancer chemotherapeutics. The paper is just a collection of information with no inference drawn. Major issues are mentioned below:
1. CBD is known to have beneficial interactions with chemotherapeutic agents and in fact that is the reason for its use clinically. The paper should be rather re-organized to highlight limited adverse drug interactions.
2. The authors have just listed the interactions with each chemotherapeutic agent with no inference.
3. All the tables have no column of key effect obtained, which is essentially required to have more understanding.
4. A section on mechanisms of drug interaction is missing.
5. Conclusions of the study is more of a summary rather than conclusion. Certain text appears like an abstract rather than concluding statements. Conclusion should rather highlight the future perspective.
6. Language needs revision.
Author Response
A point-by-point response to the reviewers
Article: Drug-drug interactions of cannabidiol with standard-of-care chemotherapeutics (ijms-2171652)
We value the reviewer's time reading our manuscript and providing valuable feedback. While revising the manuscript based on the referees' guidance, we meticulously addressed all their remarks and concerns, resulting in a better and clearer text. We hope that the reviewers will now approve the resulting revised manuscript. The point-by-point responses are given below. The yellow highlights in the revised manuscript indicate the changes. Our answers are included below.
Response to reviewer 2
The authors have discussed CBD interactions with selected anticancer chemotherapeutics. The paper is just a collection of information with no inference drawn. Major issues are mentioned below:
- CBD is known to have beneficial interactions with chemotherapeutic agents and in fact that is the reason for its use clinically. The paper should be rather re-organized to highlight limited adverse drug interactions.
We are unsure about the general applicability of the: "beneficial interactions with chemotherapeutic agents" statement. There might be a general agreement that combining cannabidiol (CBD) with some cytostatics improves anticancer therapy in terms of patients' quality of life, i.e., the reduced level of pain and toxic side effects. However, unfortunately rather little is known about whether or to what extent do such combinations impact patients' survival or disease progression – key aspects that we try to highlight in our review article as being insufficiently addressed so far. We indeed gathered and quoted here the limited information supporting the notion that combining CBD with some of the clinically used cytostatics might lead to a better therapeutic outcome in terms of survival. However, clinical data supporting such a claim are clearly limited and insufficient to reach general conclusions and recommendations. The bioactivity profile of cannabinoids is so extensive and involves so many pathways that the assumption of generally beneficial interactions may not always be valid. One can even raise the intriguing question of whether the CBD-induced lowering of the chemotherapy-caused side effects may not just reflect a decreased availability of the cytostatic drugs and hence their lower efficiency, which may, on the one hand, relieve some unpleasant effects of chemotherapy, while at the same time undermining the anticancer effects of the chemotherapy. In this regard, we even highlighted a few examples suggesting that the combined therapy might indeed attenuate the treatment outcome. We hope that by pointing out these gaps in our knowledge in this area, our review will raise precautions and motivate further research on this topic.
- The authors have just listed the interactions with each chemotherapeutic agent with no inference.
To address this point, we have included the inference aspect of the interactions in the article's final chapter (Chapter 8). In the revised manuscript, this chapter has been renamed to "Discussion," and it has been significantly extended (see lines 669-681 ).
- All the tables have no column of key effect obtained, which is essentially required to have more understanding.
That is a good point, thank you. We have added such a column (see new tables 1-5)
- A section on mechanisms of drug interaction is missing.
Such a section was present in the text; however, we agree it needed to be better highlighted and linked to the drugs. Therefore, in response to this comment, we have reformulated the relevant text (see lines: 73-90). Further mechanistic interaction insights are now also pointed out in chapters related to individual chemotherapeutics.
- Conclusions of the study is more of a summary rather than conclusion. Certain text appears like an abstract rather than concluding statements. Conclusion should rather highlight the future perspective.
As suggested by the reviewer, the future perspective has now been included in the final chapter (Chapter 8) (see lines 669-681 ).
- Language needs revision
The language has now been revised by a British company specializing in language corrections of scientific texts (www.sees-editing.co.uk). In the revised text, typing errors have also been corrected.
Reviewer 3 Report
The manuscript, Drug-drug interactions of cannabidiol with standard-of-care chemotherapeutics, is a review article that seeks to summarize current research and pre-clinical data on the interactions of CBD with various cancer therapies. The review is well written and well organized, providing a valuable resource on the topic. While the manuscript seems ready for publication in its current form, a few suggestions for improvement are included below as well.
Minor Concerns
The tables are all valuable summaries of the studies that have been done on drug interactions with CBD. However, it seems to summarize what is being tested, without comment on the result (positive or negative interaction / regulation). A very brief (one sentence) description of the outcome would significantly improve the usefulness of this chart.
The paragraph that begins on line 602 (Other modes of chemotherapy potentiation by CBD…) requires citations.
Author Response
A point-by-point response to the reviewers
Article: Drug-drug interactions of cannabidiol with standard-of-care chemotherapeutics (ijms-2171652)
We value the reviewer's time reading our manuscript and providing valuable feedback. While revising the manuscript based on the referees' guidance, we meticulously addressed all their remarks and concerns, resulting in a better and clearer text. We hope that the reviewers will now approve the resulting revised manuscript. The point-by-point responses are given below. The yellow highlights in the revised manuscript indicate the changes. Our answers are included below.
Response to reviewer 3
The manuscript, Drug-drug interactions of cannabidiol with standard-of-care chemotherapeutics, is a review article that seeks to summarize current research and preclinical data on the interactions of CBD with various cancer therapies. The review is well written and well organized, providing a valuable resource on the topic. While the manuscript seems ready for publication in its current form, a few suggestions for improvement are included below as well.
We thank the referee for positive feedback and stimulating comments. Our responses to the minor concerns are included below.
Minor Concerns
The tables are all valuable summaries of the studies that have been done on drug interactions with CBD. However, it seems to summarize what is being tested, without comment on the result (positive or negative interaction / regulation). A very brief (one sentence) description of the outcome would significantly improve the usefulness of this chart.
Thank you for the suggestion. We have added a new column addressing this point (see new tables 1-5)
The paragraph that begins on line 602 (Other modes of chemotherapy potentiation by CBD…) requires citations.
The citations have now been added to the revised manuscript (see lines 651-655)
Reviewer 4 Report
It is an important and interesting topic. Very extensive and deep review but it is easy to follow.
Only, the tables, there are some editing issues (division in two pages, etc.). Put some information as notes at the bottom on the table or use more abbreviation could help.
Author Response
A point-by-point response to the reviewers
Article: Drug-drug interactions of cannabidiol with standard-of-care chemotherapeutics (ijms-2171652)
We value the reviewer's time reading our manuscript and providing valuable feedback. While revising the manuscript based on the referees' guidance, we meticulously addressed all their remarks and concerns, resulting in a better and clearer text. We hope that the reviewers will now approve the resulting revised manuscript. The point-by-point responses are given below. The yellow highlights in the revised manuscript indicate the changes. Our answers are included below.
Response to reviewer 4
It is an important and interesting topic. Very extensive and deep review but it is easy to follow.
Only, the tables, there are some editing issues (division in two pages, etc.). Put some information as notes at the bottom on the table or use more abbreviation could help.
We thank the referee for the positive feedback. The concerns regarding tables' compactness have been addressed as suggested (see new tables 1-5).
Round 2
Reviewer 2 Report
Paper is now acceptable.